# Selenomethionine Antagonized microRNAs Involved in Apoptosis of Rat Articular Cartilage Induced by T-2 Toxin

**DOI:** 10.3390/toxins15080496

**Published:** 2023-08-04

**Authors:** Fangfang Yu, Kangting Luo, Miao Wang, Jincai Luo, Lei Sun, Shuiyuan Yu, Juan Zuo, Yanjie Wang

**Affiliations:** 1School of Public Health, Zhengzhou University, Zhengzhou 450001, China; yufangfang@zzu.edu.cn (F.Y.); lkt1307@163.com (K.L.); wangmiao_9842@163.com (M.W.); sl349496017@163.com (L.S.); ysy599926@163.com (S.Y.); zuojuan719920@163.com (J.Z.); 2Sanmenxia Center for Disease Control and Prevention, Sanmenxia 472000, China; smxwsluo@163.com

**Keywords:** T-2 toxin, selenomethionine, microRNAs, apoptosis, articular cartilage

## Abstract

T-2 toxin and selenium deficiency are considered important etiologies of Kashin–Beck disease (KBD), although the exact mechanism is still unclear. To identify differentially expressed microRNAs (DE-miRNAs) in the articular cartilage of rats exposed to T-2 toxin and selenomethionine (SeMet) supplementation, thirty-six 4-week-old Sprague Dawley rats were divided into a control group (gavaged with 4% anhydrous ethanol), a T-2 group (gavaged with 100 ng/g·bw/day T-2 toxin), and a T-2 + SeMet group (gavaged with 100 ng/g·bw/day T-2 toxin and 0.5 mg/kg·bw/day SeMet), respectively. Toluidine blue staining was performed to detect the pathological changes of articular cartilage. Three rats per group were randomly selected for high-throughput sequencing of articular cartilage. Target genes of DE-miRNAs were predicted using miRanda and RNAhybrid databases, and the Gene Ontology and Kyoto Encyclopedia of Genes and Genomes pathway were enriched. The network map of miRNA-target genes was constructed using Cytoscape software. The expression profiles of miRNAs associated with KBD were obtained from the Gene Expression Omnibus database. Additionally, the DE-miRNAs were selected for real-time quantitative PCR (RT-qPCR) verification. Toluidine blue staining demonstrated that T-2 toxin damaged articular cartilage and SeMet effectively alleviated articular cartilage lesions. A total of 50 DE-miRNAs (28 upregulated and 22 downregulated) in the T-2 group vs. the control group, 18 DE-miRNAs (6 upregulated and 12 downregulated) in the T-2 + SeMet group vs. the control group, and 25 DE-miRNAs (5 upregulated and 20 downregulated) in the T-2 + SeMet group vs. the T-2 group were identified. Enrichment analysis showed the target genes of DE-miRNAs were associated with apoptosis, and in the MAPK and TGF-β signaling pathways in the T-2 group vs. the control group. However, the pathway of apoptosis was not significant in the T-2 + SeMet group vs. the control group. These results indicated that T-2 toxin induced apoptosis, whereas SeMet supplementation antagonized apoptosis. Apoptosis and autophagy occurred simultaneously in the T-2 + SeMet group vs. T-2 group, and autophagy may inhibit apoptosis to protect cartilage. Compared with the GSE186593 dataset, the evidence of miR-133a-3p involved in apoptosis was more abundant. The results of RT-qPCR validation were consistent with RNA sequencing results. Our findings suggested that apoptosis was involved in articular cartilage lesions induced by T-2 toxin, whereas SeMet supplementation antagonized apoptosis, and that miR-133a-3p most probably played a central role in the apoptosis process.

## 1. Introduction

The T-2 toxin, a cytotoxic fungal secondary metabolite, is derived from *Fusarium oxysporum*, and it is widely distributed in cereals and animal feed [1]. The T-2 toxin causes harm to humans and animals, as well as damages to the nervous system, immune system, digestive system, reproductive system, and other systems [2,3,4,5,6]. Kashin–Beck disease (KBD) is an endemic and chronic osteoarthritis, and the clinical changes include thickening and deformities of limbs, short fingers, short limbs and dwarfism, etc. The pathological changes are primarily degeneration and necrosis of articular cartilage and epiphyseal plate cartilage [7]. Although more than 50 environmental risk factors for KBD have been identified, the etiology and pathology of the disease have not yet been conclusively determined [8]. The T-2 toxin is considered to be one of the important etiologies of KBD. In an epidemiological survey, T-2 toxin was measured more frequently in KBD patients [9]. Furthermore, the T-2 toxin can damage human articular chondrocytes through degrading the extracellular matrix [10,11,12]. However, the potential mechanism of T-2-induced articular cartilage damage is unclear.

Selenium (Se) deficiency is also considered as a primary etiological hypothesis for KBD [13]. Se is an essential trace element required for the growth and development for humans and animals [14]. Se deficiency not only damages multiple tissues and organs in animals, such as the liver, spleen, and testis [15,16], but also causes the decline of human immunity and an increase in cancer risk [17,18]. Se occurs in both inorganic and organic forms, and organic Se, such as selenomethionine (SeMet), is easily absorbed and utilized and frequently used as a Se supplement [19]. Previous studies confirmed that the T-2 toxin combined with Se deficiency may be the cause of KBD. In rat models, increased levels of FGF3 and FGFR2 were observed in the cartilage with the T-2 toxin plus low Se diets [20]. With combined low Se and T-2 toxin, it was discovered the levels of heat shock protein 47 and type II collagen decreased significantly, which increased cartilage degradation and cartilage damage [11]. Additionally, supplementing Se in KBD endemic areas had a certain effect on the prevention and treatment of children with KBD [21].

MicroRNAs (miRNAs) are a class of 18–23 nucleotides of short-stranded non-coding RNA which cannot encode proteins [22]. miRNAs can inhibit the translation of mRNAs or directly degrade mRNAs by binding to the 3’ untranslated region of a specific target gene mRNA [23]. miRNAs play crucial roles in the processes of cell differentiation, apoptosis, tumor formation, viral replication, and tissue development [24,25,26]. Previous studies have proved that the abnormal expression of miRNA caused various human diseases, including cancer, cardiovascular disease, and diabetes [27,28,29]. Therefore, miRNAs have been proposed as biomarkers for early detection and predictive diagnosis of diseases. miRNAs are closely related to cartilage, such as miR-410-3a, miR-18a-3p, miR-369-3p, and miR-206 that were involved in cartilage lesions. It had been reported that miR-410-3p was significantly downregulated in the articular cartilage of mice osteoarthritis and regulated apoptosis and inflammation of chondrocytes through the NF-κB signal pathway [30]. miR-18a-3p was upregulated in chondrocytes treated with IL-1β and caused apoptosis. After knocking down miR-18a-3p, the degree of cell apoptosis was reduced [31]. The expression of miR-369-3p was significantly increased in cartilage tissues, targeting JAK2 to induce apoptosis of chondrocytes [32]. The level of miR-206 was significantly increased in human chondrocytes, which inhibited chondrocyte proliferation and promoted apoptosis [33]. However, articular cartilage lesions induced by the T-2 toxin has not yet been fully understood, and few investigations have been conducted on the miRNA expression patterns of articular cartilage lesions induced by T-2 toxin. The objective of this study was to identify the differentially expressed miRNAs (DE-miRNAs) in articular cartilage lesions induced by the T-2 toxin, as well as the alteration of miRNAs after SeMet supplementation, and provided a theoretical basis for understanding the role of miRNA in the action of T-2 toxin and SeMet in articular cartilage.

## 2. Results

### 2.1. Toluidine Blue Staining

The toluidine blue staining of articular cartilage among different groups was shown in Figure 1. In the control group, the cartilage matrix was deep-purple and the chondrocytes were closely arranged. The cartilage matrix in the T-2 group was less colored and the number of chondrocytes was less than that in the control group. Compared with the T-2 group, the cartilage matrix in the T-2 + SeMet group was more stained, and the number of chondrocytes increased. These results suggested that the T-2 toxin degraded the cartilage matrix and reduced the number of chondrocytes, causing articular cartilage lesions, and SeMet effectively alleviated articular cartilage lesions.

### 2.2. Sequencing Quality Analysis

The Sequencing data quality was assessed using FastQC software. Quality control standard was Q30% > 85% and CG% in the range of 40–60%. As shown in Table 1, the Q30% ranged from 96.4% to 97.5%, and the CG% ranged from 46.1% to 48.0%, indicating the sequencing quality was good. Small RNA fragments were mainly in the length range of 18–26 nucleotides, with a peak of 22 nucleotides (Appendix A). This was consistent with the characteristics of miRNA length distribution and indicated the quality of sequencing data was reliable.

### 2.3. Screening for DE-miRNAs

A total of 2605 miRNAs (763 known miRNAs and 1842 novel miRNAs) were determined in the constructed small RNA library. Compared with the control group, 50 DE-miRNAs were identified in the T-2 group (28 upregulated and 22 downregulated), and 18 miRNAs were differentially expressed (6 upregulated and 12 downregulated) in the T-2 + SeMet group. Compared with the T-2 group, 25 miRNAs were differentially expressed (5 upregulated and 20 downregulated) in the T-2 + SeMet group. In the volcano plot (Figure 2A–C), the red and green dots represent upregulated and downregulated miRNAs, respectively. The black dots represent miRNAs with non-significant differential expression. In Figure 2D–F, the heatmap shows the DE-miRNAs between-group variation and within-group variation.

### 2.4. Functional Enrichment Analysis of Target Genes

A total of 870, 170, and 595 target genes were predicted in the T-2 group vs. the control group, the T-2 + SeMet group vs. the control group, and the T-2 + SeMet group vs. the T-2 group, respectively. The Gene ontology (GO) analysis and Kyoto Encyclopedia of Genes and Genomes (KEGG) enrichment analysis were performed on the predicted target genes. In the T-2 group vs. the control group, GO analysis indicated the biological process mainly included the negative regulation of T cell proliferation and intrinsic apoptotic signaling pathway in response to DNA damage. The cellular component included membrane and nuclear. Furthermore, the molecular function was related to mitogen-activated protein kinase binding, Ral GTPase binding, and cyclin-dependent protein kinase activity. In the T-2 + SeMet group vs. the control group, the biological process included G1 DNA damage checkpoint and glomerular epithelial cell differentiation. The cellular component included the intracellular component and the membrane, and the molecular function mainly included extracellular matrix binding and Hsp90 protein binding. In the T-2 + SeMet group vs. the T-2 group, the biological process included the negative regulation of cellular catabolic process, the cellular component included the extracellular matrix and the membrane, and the molecular function mainly included complement binding and core promoter sequence specific DNA binding (Figure 3).

KEGG enrichment analysis identified pathways that were significantly enriched in apoptosis, and the MAPK and TGF-β signaling pathway in the T-2 group vs. the control group, of which the apoptosis pathway was the most significant (Figure 4A). The chemokine signaling pathway, growth hormone synthesis, and secretion and action pathway were enriched in the T-2 + SeMet group vs. the control group, but no apoptosis pathway was observed (Figure 4B). The apoptosis and autophagy pathway appeared simultaneously in the T-2 + SeMet group vs. the T-2 group (Figure 4C).

### 2.5. Apoptosis-Related Target Genes and miRNAs

The target genes enriched in the apoptosis pathway and their corresponding miRNAs were shown in Table 2 and Table 3. The overlapping target genes and DE-miRNAs associated with apoptosis were shown in Figure 5A,B. To understand the target genes (Rela, Bbc3 and Mapk1) corresponding to miR-133a-3p, miR-675-5p, and miR-667-3p, the expression of three target genes was identified, and their localization in cells was positioned according to the Human Protein Atlas (HPA) database. Immunohistochemistry analysis showed that the protein expression level of Rela in skeletal muscle was lower than that in bone marrow, the level of Mapk1 in skeletal muscle was higher than that in bone marrow, and the level of Bbc3 in bone marrow was similar to that in skeletal muscle. The subcellular map showed that Rela and Bbc3 were located in the cytosol. Mapk1 was mostly localized in the cytosol, and additional locations were found in nuclear speckles (Figure 5C).

### 2.6. Regulatory Network of miRNA and Its Target Genes

To reveal the relationship between DE-miRNAs and target genes, the miRNA–mRNA integrated analysis was carried out. In this analysis, we selected nine differentially co-expressed miRNAs associated with apoptosis in the T-2 group vs. the control group and the T-2 + SeMet group vs. the T-2 group, with 203 target genes (Figure 6). Most DE-miRNAs had more than five target genes. For example, rno-miR-18a-3p was correlated with 78 target genes, of which one target gene was associated with other DE-miRNAs.

### 2.7. miRNA Expression Profiles Related to KBD

To demonstrate that the sequencing results of this study were reliable, the Gene Expression Omnibus (GEO) dataset related to KBD was retrieved. A total of 1550 DE-miRNAs were identified in the GSE186593 dataset, and 868 of these were upregulated, whereas 682 were downregulated. The DE-miRNAs of the GSE186593 dataset were compared with the DE-miRNAs of this study, and 15 overlapping DE-miRNAs were identified (Table 4). As shown in Figure 7, miR-133a-3p was co-expressed in the T-2 group vs. the control group, the T-2 + SeMet group vs. the T-2 group, and the KBD vs. the normal group, and was also related to the apoptosis pathway.

### 2.8. Validation of DE-miRNAs

To validate the reliability of RNA sequencing, three DE-miRNAs were randomly selected from the T-2 group vs. the control group (miR-206-3p) and the T-2 + SeMet group vs. the T-2 group (miR-204-5p, miR-376c-5p) for real-time quantitative PCR (RT-qPCR) verification. The trends of variation in DE-miRNAs were consistent with RNA sequencing (Figure 8).

## 3. Discussion

T-2 toxin and Se deficiency have been widely studied as the two main causes of KBD [8]. Emerging evidence has shown the important function of miRNAs in osteoarticular diseases, and how miRNAs can be used as important biomarkers for early diagnosis and potential molecular targets for treatment [34]. However, few studies have focused on miRNA expression analysis of articular cartilage lesions induced by the T-2 toxin and treated with SeMet supplementation. A study found that miR-140 deficiency aggravated T-2 toxin-induced knee cartilage injury in mice, which may be involved in extracellular matrix degradation [35]. However, more research is needed on the specific mechanism of miRNA in T-2 toxin-induced cartilage lesions and the role of selenium supplements.

We identified 50, 18, and 25 DE-miRNAs in the T-2 group vs. the control group, the T-2 + SeMet group vs. the control group, and the T-2 + SeMet group vs. the T-2 group, respectively. These DE-miRNAs may reflect the degree of articular cartilage lesion, indirectly indicating that the articular cartilage was seriously damaged when the rats were treated with the T-2 toxin alone, and that this was alleviated with SeMet supplementation. In addition, we found that the expression of miRNAs with significant differences in the T-2 group vs. the control group was the opposite of those in the T-2 + SeMet group vs. the T-2 group. This evidence suggested that miRNA expression participated in the formation and resolution of lesions in articular cartilage. To understand the role of DE-miRNAs in articular cartilage with the T-2 toxin and SeMet, functional enrichment analysis was performed. KEGG pathway enrichment analysis demonstrated that DE-miRNAs were mainly enriched in the apoptosis pathway and the MAPK and TGF-β signaling pathways in the T-2 group compared with the control group. Apoptosis occurred in chondrocytes exposed to the T-2 toxin, which was consistent with previous studies [36,37]. T-2 toxin induced cartilage lesions in rats, and the levels of caspase-3, p53, Bax protein, and mRNA were higher than those in the control group, while the Bcl-2 level was lower, indicating that the T-2 toxin induced chondrocyte apoptosis [38]. As we all know, apoptosis is a type of programmed cell death, that is important in the growth and development of the organism to maintain its metabolism and remove abnormal cells [39]. The T-2 toxin inhibited protein synthesis, caused oxidative stress, and induced chondrocyte apoptosis through mitochondrial-mediated pathways [40]. Furthermore, the T-2 toxin indirectly activated several signal pathways and led to apoptosis of chondrocytes. MAPK is a member of the serine/threonine protein kinase family that can regulate the expression of various genes [41]. The MAPK signaling pathway has been widely confirmed to be involved in chondrocyte-related processes, including apoptosis, cell differentiation, and inflammation [42]. In particular, the T-2 toxin induced apoptosis of human neuronal cells through the P38 MAPK pathway [43]. Additionally, the TGF-β signaling pathway regulates the proliferation, differentiation, and apoptosis of chondrocytes [44]. TGF-β1 is a member of TGF-β superfamily, and its overexpression alleviated the inhibitory effect of miR-296-5p on chondrocyte apoptosis suggested that TGF-β1 involved chondrocyte apoptosis [45].

However, the apoptosis pathway was not significantly enriched in the T-2 + SeMet group vs. the control group. Combined with the KEGG enrichment analysis results of the T-2 group vs. the control group, it was shown that the T-2 toxin could induce apoptosis, whereas SeMet indirectly antagonized apoptosis. Oxidative stress and apoptosis have been shown to occur in mice exposed to the T-2 toxin, and apoptosis was alleviated after treatment with Se [46]. Moreover, the apoptosis pathway was significantly enriched in the T-2 + SeMet group vs. the T-2 group, and the autophagy pathway appeared simultaneously. Autophagy is a process of cell self-degradation that plays an important role in intracellular environmental stability and protects chondrocytes [47]. Although significant differences in metabolism and morphology were observed between apoptosis and autophagy, accumulating evidence shows that they are closely related to the occurrence and reaction of chondrocyte lesions. Moderate autophagy has been reported to inhibit apoptosis [48]. By contrast, overactivated autophagy can cause cell death by activating pre-apoptotic factors in mitochondria [49]. The cytoprotective function of autophagy is regulated by the negative regulation of apoptosis, and apoptosis signal transduction subsequently inhibits autophagy [50]. Therefore, we consider that autophagy played a certain role in the process of SeMet alleviation of apoptosis in this study. However, the effect of SeMet did not completely inhibit the occurrence of apoptosis. Overall, our results demonstrated that the T-2 toxin induced cartilage apoptosis, whereas SeMet antagonized apoptosis.

The five overlapping target genes (Bbc3, Rela, Mapk1, Ntrk1, and Htra2) and three overlapping miRNAs (miR-133a-3p, miR-675-5p, and miR-667-3p) were identified in the two apoptosis pathways. It was suggested these DE-miRNAs may be involved in the process of cartilage apoptosis. The expression level of the 3 DE-miRNAs were upregulated in the T-2 group vs. the control group, and these were downregulated in the T-2 + SeMet group vs. the T-2 group, so it was probable that their abnormal expression was involved in the occurrence of apoptosis. The expression of miR-675-5p was inhibited after strontium ranelate treatment in mice, and the overexpression of miR-675-5p decreased the proliferation of skeletal muscle cells and promoted cell apoptosis [51]. miR-133a-3p inhibited cell proliferation, invasion, and migration by targeting COL1A1, and promoted apoptosis of Esophageal Squamous Cell Carcinoma [52]. Moreover, long noncoding RNA LINC01278 could inhibit miRNA-133a-3p and promote the proliferation of osteosarcoma cells and inhibit apoptosis [53]. These previous reports were consistent with our study, where overexpression of miR-133a-3p and miR-675-5p promoted apoptosis, but where apoptosis was inhibited when the expression of these miRNAs were downregulated. By comparing it with the GSE186593 dataset, only miR-133a-3p was associated with apoptosis, indicating that this was the most likely miRNA to participate in the apoptosis of articular cartilage.

However, this study has limited information on the molecular mechanisms involving miRNAs in the induction of articular cartilage by the T-2 toxin and SeMet supplementation. To evaluate the role of specific DE-miRNAs in the induction of articular cartilage by the T-2 toxin, we intend to explore the potential molecular mechanisms of articular cartilage induced by the T-2 toxin and SeMet in future experiments.

## 4. Conclusions

Apoptosis was involved in the process of articular cartilage lesions induced by the T-2 toxin, and that apoptosis was antagonized by SeMet supplementation. Moreover, miR-133a-3p, miR-675-5p, and miR-667-3p may play an important role in apoptosis. In particular, the evidence for miR-133a-3p involvement in this process is stronger. These results suggested a miRNA-related mechanism of articular cartilage lesions induced by the T-2 toxin and the mitigation by SeMet.

## 5. Materials and Methods

### 5.1. Experimental Animal and Tissue Extraction

Thirty-six healthy male, specific-pathogen-free (SPF), Sprague Dawley rats (4 weeks old, weighing 60–80 g) were selected and purchased from the Henan provincial laboratory animal center. The rats were acclimatized and fed for 1 week, then randomly divided into 3 groups (control group, T-2 group, and T-2 + SeMet group). The rats in the control group were given 4% anhydrous ethanol by gavage, the rats in the T-2 group were given 100 ng/g·bw/day T-2 toxin, and the rats in the T-2 + SeMet group were given 100 ng/g·bw/day T-2 toxin and 0.5 mg/kg·bw/day SeMet. The intervention doses were chosen based on previous studies [54,55]. After 4 weeks continuous administration, three rats in each group were randomly selected, and the articular cartilage was taken after anesthesia with 7% chloral hydrate. All animal studies were reviewed and approved by the animal ethics research committee of Zhengzhou University.

### 5.2. Toluidine Blue Staining

Articular cartilage was decalcified, embedded in paraffin wax, sliced, dewaxed with xylene twice, and dewaxed with gradient ethanol. The sections were stained with toluidine blue for 30 min, and then washed with tap water. Using acetone differentiation section, the chondrocytes were deep-blue and clearly visible and then dehydrated in graded ethanol. Xylene was dropped on the slice to make the slice transparent, and rhamsan gum was finally added for sealing. The cartilage lesions of articular cartilage in the rats in each group were observed by microscope.

### 5.3. RNA Extraction

The total RNA of articular cartilage was extracted using Trizol (Thermo Fisher Scientific, Waltham, MA, USA). The concentration of the extracted RNA sample was quantified using Qubit (sample concentration ≥ 200 ng/µL, total ≥ 2 μg), and the RNA purity was measured using a Nanodrop 2000 spectrophotometer (Thermo Fisher Scientific, Waltham, MA, USA). The 260/280 value was 1.8–2.2, the 260/230 value was ≥ 2.0. The RNA degradation and contamination were analyzed by agarose gel electrophoresis, and then the integrity of the RNA (RNA integrity number ≥ 7) was measured using the Agilent 2100 Bioanalyzer (Agilent Technologies, Santa Clara, CA, USA).

### 5.4. Library Construction and RNA Sequencing

The library construction and high-throughput sequencing were performed by Shanghai Tianhao Biotechnology Co. Total RNA was used for the splice ligation reaction, and one strand of cDNA was synthesized by reverse transcription using SuperScript IV reverse transcriptase. A double-stranded library of miRNA was synthesized and amplified using PCR, and polyacrylamide gel electrophoresis was used to recover and purify the approximately 140–160 bp bands. Finally, the Agilent 2100 Bioanalyzer was used to control the quality of the library and sequenced using Illumina Hiseq2500 (Illumina, San Diego, CA, USA).

### 5.5. Quality Assessment of Raw Sequencing Data 

Raw reads obtained from sequencing contained some low-quality and junctional reads. To ensure the quality of information analysis, the raw reads were filtered using FastQC software (http://www.bioinformatics.babraham.ac.uk/projects/fastqc/ (accessed on 13 August 2022)), to obtain clean reads for subsequent analysis. The filtering steps used were as follows: (1) removing the splice sequences and sequences without a 3’ splice and insert fragment; (2) removing sequences with Q20 ratios < 60%; (3) removing sequences with length ranges outside the 18–36 bp range.

### 5.6. Identification of DE-miRNAs

The small RNA sequences were compared with the miRBase database (https://www.mirbase.org/ (accessed on 13 August 2022)) to identify both known miRNAs and novel miRNAs. The DE-miRNAs were analyzed using Deseq2 software (https://support.bioconductor.org/ (accessed on 13 August 2022)). The screening threshold of *p* < 0.05 and |log_2_(fold change)| > 1 were considered as DE-miRNAs, where log_2_(fold change) > 1 was marked as upregulated DE-miRNAs and log_2_(fold change) < −1 was marked as downregulated DE-miRNAs.

### 5.7. miRNA Target Gene Prediction and Functional Enrichment Analysis

The miRNA regulates the expression of target genes by specifically binding to their mRNA. To improve the accuracy of the results, miRanda and RNAhybrid software (http://www.microrna.org/microrna/home.do (accessed on 13 August 2022), http: //bibiserv.techfak.uni-bielefeld.de/rnahybrid/ (accessed on 13 August 2022)) were used to predict the target genes of identified miRNA, and the intersection of the two methods was the target of DE-miRNAs. The biological functions and pathways involved in DE-miRNAs were predicted using Gene ontology (GO) analysis and Kyoto encyclopedia of genes and genomes (KEGG) analysis.

### 5.8. Regulatory Network of miRNAs-Target Genes

According to the targeting-binding relationship between miRNAs and target genes, a regulatory network was constructed and visualized using Cytoscape software (https://cytoscape.org/ (accessed on 20 May 2023)).

### 5.9. Database Verification of DE-miRNAs

The Gene Expression Omnibus (GEO, https://www.ncbi.nlm.nih.gov/geo/ (accessed on 20 May 2023)) database is the largest fully open high-throughput molecular information database under the platform of the National Biotechnology Information Center (NCBI). The dataset GSE186593 of miRNA expression profiling related to KBD was downloaded from the GEO database. GSE186593 gene chip data detection platform was GPL16791 IlluminaHiSeq2500 (*Homo sapiens*). The subjects of the dataset were five healthy individuals and five patients with KBD with sampling of elbow venous blood.

### 5.10. Verification of miRNAs and Target Gene Analysis

Total RNA was extracted from the articular cartilage of each group (*n* = 3) using Trizol reagent (Invitrogen, Waltham, MA, USA) according to manufacturer’s protocol and reverse transcribed into cDNA using TransScript FirstStrand cDNA Synthesis Kit following the manufacturers’ instructions. The relative miRNA expression levels were determined using real-time quantitative polymerase chain reaction (RT-qPCR) The relative expression levels of miRNA were calculated using the 2^−ΔΔCt^ method (U6 was selected as internal reference). The primers used qPCR analysis were shown in Appendix A. The Human Protein Atlas (HPA, https://www.proteinatlas.org/ (accessed on 20 May 2023)) database is dedicated to providing tissue and cell distribution information of all 24,000 human proteins, which is available to the public free of charge. The proteins expression levels and location of the target genes in the cells were obtained from the HPA database.

## Figures and Tables

**Figure 1 toxins-15-00496-f001:**
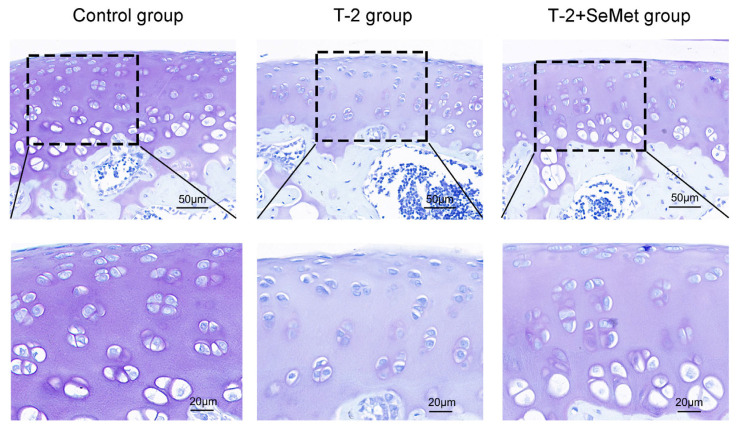
Toluidine blue staining among different groups. The nucleus of the chondrocytes was deep-blue and the cartilage matrix was purple. The black dashed box represents partially enlarged area.

**Figure 2 toxins-15-00496-f002:**
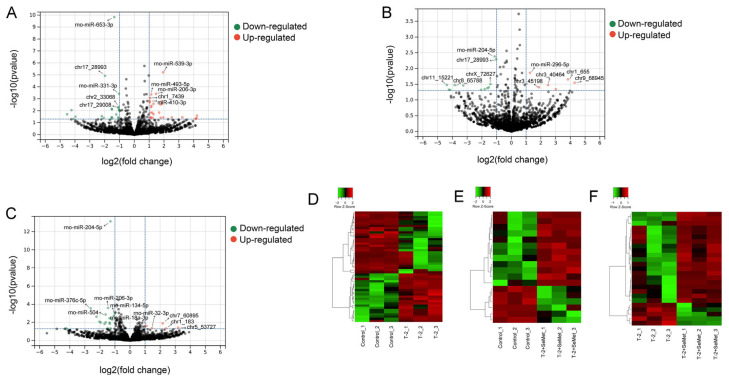
The differentially expressed miRNAs (DE-miRNAs) in articular cartilage of rats. Volcano plots of DE-miRNAs are compared between the T-2 group and the control group (**A**), between the T-2 + SeMet group and the control group (**B**), and between the T-2 + SeMet group and the T-2 group (**C**). Heatmap of DE-miRNAs in the T-2 group vs. the control group (**D**), the T-2 + SeMet group vs. the control group (**E**) and the T-2 + SeMet group vs. the T-2 group (**F**). The color scale-bar from green to red indicates increased expression levels of miRNA. The threshold of significance was *p* < 0.05, *n* = 3.

**Figure 3 toxins-15-00496-f003:**
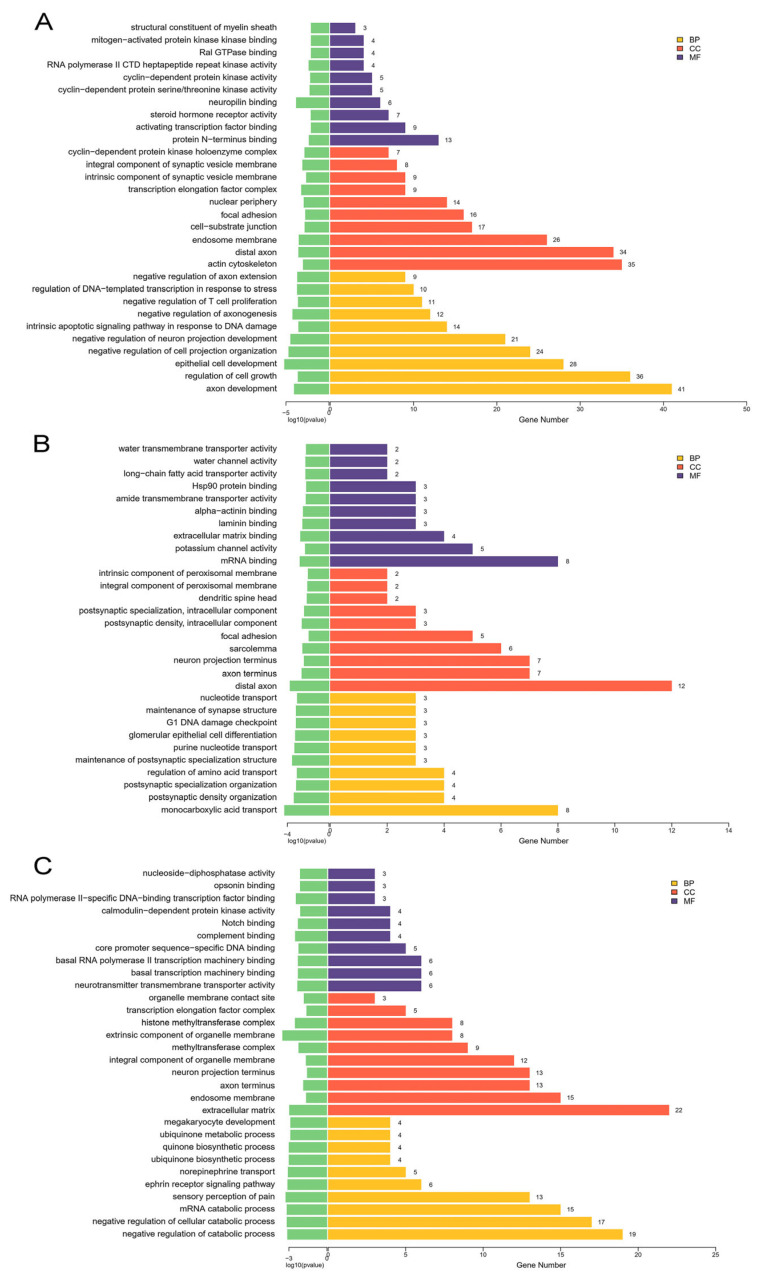
GO analysis of target genes. The top 30 GO terms of target genes in the T-2 group vs. the control group (**A**), the T-2 + SeMet group vs. the control group (**B**) and the T-2 + SeMet group vs. the T-2 group (**C**). The BP represents biological process, CC represents cellular component, and MF represents molecular function. The threshold of significance was *p* < 0.05, *n* = 3.

**Figure 4 toxins-15-00496-f004:**
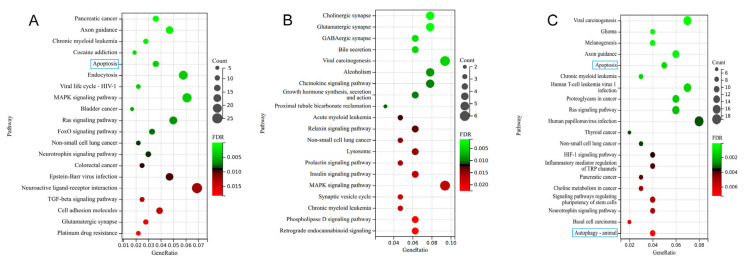
KEGG enrichment analysis of target genes. (**A**–**C**) indicate the top 20 enriched KEGG pathways of target genes in the T-2 group vs. the control group, the T-2 + SeMet group vs. the control group, and the T-2 + SeMet group vs. the T-2 group, respectively. The blue boxes represent apoptosis and autophagy pathways. The threshold of significance was *p* < 0.05, *n* = 3.

**Figure 5 toxins-15-00496-f005:**
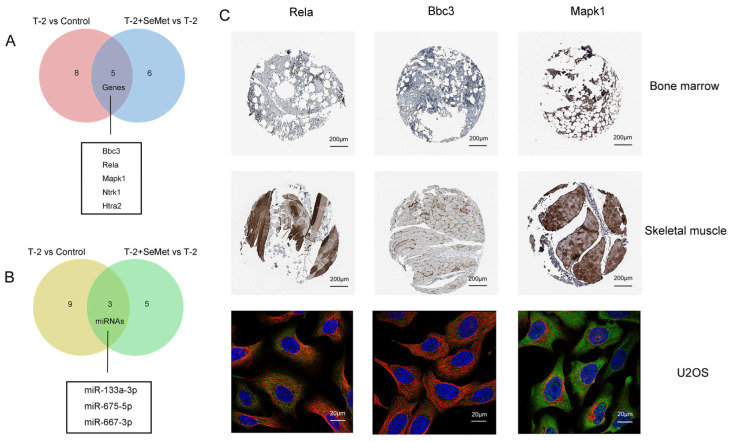
The Venn diagram displays the overlapping target genes and DE-miRNAs associated with apoptosis in the T-2 group vs. the control group (**A**) and the T-2 + SeMet group vs. the T-2 group (**B**). The threshold of significance was *p* < 0.05, *n* = 3. The immunohistochemical map shows the expression of proteins in the bone marrow and skeletal muscle, and the subcellular map shows the specific location of the proteins in U2OS cells (**C**).

**Figure 6 toxins-15-00496-f006:**
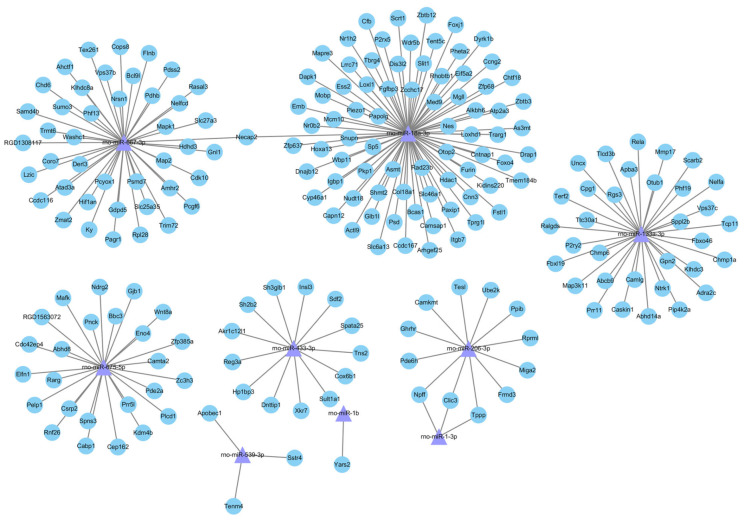
The network of apoptosis-related target genes and their related DE-miRNAs. Circles represent target genes and triangles represent DE-miRNAs. The threshold of significance was *p* < 0.05, *n* = 3.

**Figure 7 toxins-15-00496-f007:**
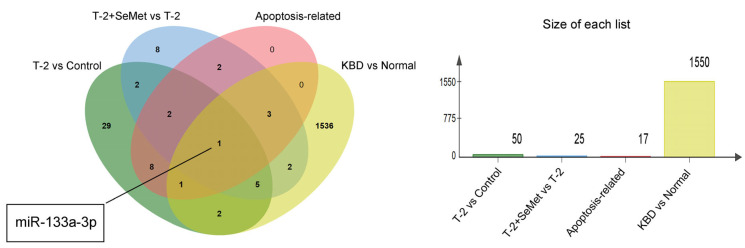
The Venn diagram displays the overlapping DE-miRNAs in the T-2 group vs. the control group, the T-2 + SeMet group vs. the T-2 group, and the KBD vs. the normal and vs. the apoptosis-related.

**Figure 8 toxins-15-00496-f008:**
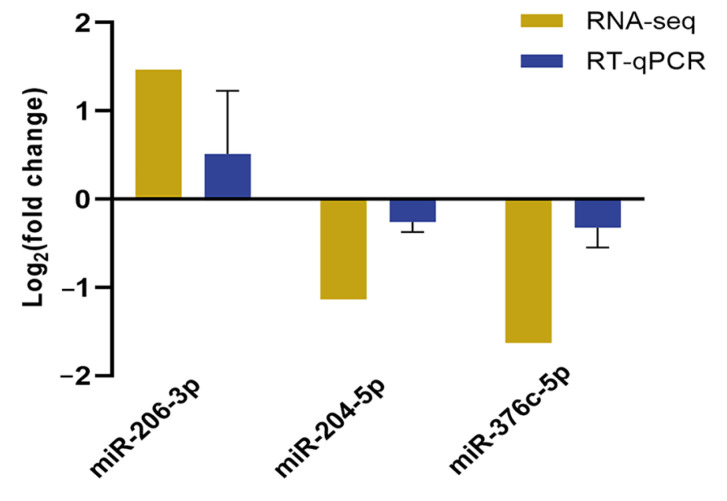
The expression of the three DE-miRNAs identified by RNA sequencing was analyzed using RT-qPCR. Mean ± SEM, *n* = 3, normalized to U6. The *x*-axis represents the miRNAs and the *y*-axis represents the log_2_ (fold change).

**Table 1 toxins-15-00496-t001:** The results of sequencing data quality control.

Sample	Read Sum	Base Sum	Q20 (%)	Q30 (%)	CG (%)
T-2_1	10,127,903	217,651,459	99.3%	97.5%	48.0%
T-2_2	10,686,280	229,331,997	98.9%	96.4%	46.1%
T-2_3	10,613,925	227,436,532	99.3%	97.4%	47.3%
T-2 + SeMet_1	10,634,792	229,547,141	99.3%	97.3%	46.5%
T-2 + SeMet_2	11,988,000	258,576,545	99.3%	97.3%	46.8%
T-2 + SeMet_3	10,488,874	225,625,339	99.2%	97.0%	46.8%
Control_1	13,241,530	284,432,884	99.3%	97.3%	47.7%
Control_2	10,752,588	234,037,192	99.2%	97.0%	46.2%
Control_3	10,914,184	235,810,258	99.3%	97.2%	47.0%

**Table 2 toxins-15-00496-t002:** The apoptosis pathway enriched target genes and corresponding differentially expressed miRNAs in the T-2 group vs. the control group.

Gene Symbol	miRNA	Log_2_FC	*p*	Type
Lmnb2	miR-489-3p	−1.54	4.90 × 10^−2^	Down
Dffa	miR-466c-3p	−1.08	3.60 × 10^−2^	Down
Traf1	miR-3577	−3.96	3.09 × 10^−2^	Down
Jun	miR-3577	−3.96	3.09 × 10^−2^	Down
Birc5	miR-3577	−3.96	3.09 × 10^−2^	Down
Bcl2l11	miR-466b-2-3p	−1.02	8.36 × 10^−3^	Down
Bcl2l11	miR-466b-4-3p	−1.02	8.36 × 10^−3^	Down
Bbc3	miR-331-3p	−1.03	3.80 × 10^−4^	Down
Bbc3	miR-675-5p	1.84	1.68 × 10^−3^	Up
Ntrk1	miR-133a-3p	1.11	3.45 × 10^−2^	Up
Ntrk1	miR-133b-3p	1.21	1.49 × 10^−2^	Up
Rela	miR-133a-3p	1.11	3.45 × 10^−2^	Up
Rela	miR-133b-3p	1.21	1.49 × 10^−2^	Up
Mapk1	miR-667-3p	1.29	1.98 × 10^−2^	Up
Gadd45b	miR-483-3p	1.68	1.43 × 10^−2^	Up
Htra2	miR-483-3p	1.68	1.43 × 10^−2^	Up
Bad	miR-540-5p	1.15	3.82 × 10^−3^	Up

**Table 3 toxins-15-00496-t003:** The apoptosis pathway enriched target genes and corresponding differentially expressed miRNAs in the T-2 + SeMet group vs. the T-2 group.

Gene Symbol	miRNA	Log2FC	*p*	Type
Ntrk1	miR-133a-3p	−1.36	1.09 × 10^−2^	Down
Rela	miR-133a-3p	−1.36	1.09 × 10^−2^	Down
Map3k14	miR-134-5p	−1.00	1.01 × 10^−3^	Down
Pik3r2	miR-134-5p	−1.00	1.01 × 10^−3^	Down
Ctsh	miR-204-5p	−1.30	7.31 × 10^−14^	Down
Bak1	miR-204-3p	−2.23	2.36 × 10^−3^	Down
Nras	miR-204-3p	−2.23	2.36 × 10^−3^	Down
Htra2	miR-668	−1.29	4.62 × 10^−2^	Down
Bbc3	miR-675-5p	−1.34	1.39 × 10^−2^	Down
Mapk1	miR-667-3p	−1.69	1.44 × 10^−2^	Down
Ctsb	miR-3099	−1.48	4.40 × 10^−2^	Down

**Table 4 toxins-15-00496-t004:** Overlapping differentially expressed miRNAs between this study and the GSE186593 dataset.

miRNAs	T-2 vs. Control		T-2 + SeMet vs. Control		T-2 + SeMet vs. T-2		KBD vs. Normal	
	*p*	Log_2_FC	*p*	Log_2_FC	*p*	Log_2_FC	*p*	Log_2_FC
miR-539-3p	6.08 × 10^−6^	1.94			3.92 × 10^−3^	−1.23	6.59 × 10^−10^	4.62
miR-1-3p	2.64 × 10^−3^	1.84			8.55 × 10^−3^	−2.03	8.59 × 10^−32^	4.63
miR-433-3p	2.60 × 10^−3^	1.30			4.96 × 10^−3^	−1.19	1.43 × 10^−26^	6.49
miR-133a-3p	3.45 × 10^−2^	1.11			1.09 × 10^−2^	−1.36	1.31 × 10^−5^	2.93
miR-410-3p	1.59 × 10^−3^	1.03			1.78 × 10^−3^	−1.07	1.23 × 10^−5^	2.91
miR-18a-3p	2.06 × 10^−2^	−1.21			2.40 × 10^−2^	1.09	1.75 × 10^−39^	−2.74
miR-483-3p	1.44 × 10^−2^	1.68					1.68 × 10^−3^	−6.10
miR-493-5p	4.37 × 10^−4^	1.14					4.18 × 10^−24^	5.94
miR-369-3p	1.92 × 10^−3^	1.05					3.59 × 10^−6^	3.05
miR-296-5p			1.37 × 10^−2^	1.26			5.05 × 10^−17^	−4.91
miR-204-5p			4.18 × 10^−3^	−1.14	7.31 × 10^−14^	−1.30	1.46 × 10^−17^	4.81
miR-376c-5p			4.20 × 10^−2^	−1.63	2.81 × 10^−4^	−2.61	7.62 × 10^−3^	4.94
miR-32-3p					4.27 × 10^−2^	1.46	2.48 × 10^−26^	2.93
miR-204-3p					2.36 × 10^−3^	−2.23	1.54 × 10^−2^	4.68
miR-134-5p					1.01 × 10^−3^	−1.00	4.28 × 10^−11^	3.75

## Data Availability

Not applicable.

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
