# Peer review of "Selenomethionine Antagonized microRNAs Involved in Apoptosis of Rat Articular Cartilage Induced by T-2 Toxin"

_toxins, 2023, doi:10.3390/toxins15080496_

Round 1

Reviewer 1 Report

1.         The text on line 136 expresses “To understand the target genes (Rela, Bbc3 and Mapk1) corresponding to miR-133a-3p, miR-675-5p and miR-667-3p, the expression of 3 target genes were identified.”, But according to Tables 1 and 2, Ntrk1 is also the target gene of miR-133a-3p. Why did this gene not be detected in subsequent immunohistochemistry?

2.         miR-133a-3p, miR-675-5p, miR-667-3p and their target genes were obtained from articular cartilage. Why is immunohistochemical validation of target genes performed in bone marrow and skeletal muscle?

3.         Results 2.2, 2.3 and 2.4 are all based on the pairwise comparison of the three treatment groups. I regard that it is more important to find out the differential genes through comprehensive comparison of the three groups, and then carry out biological function analysis and target genes prediction.

4.         What is the source of biological samples about the dataset GSE186593352 of miRNAs expression profiling related to KBD that were downloaded from GEO database? The author did not explain this key issue. If the dataset GSE186593352 is sourced from the biological samples of non SD rates, then the two kinds of data are not comparable.

5.         Please mark out the most significantly expressed differentially expressed genes in the volcanic map of Figure 1, such as the top ten.

6.         The statistical difference between the histogram in Figure 5 and Figure 6 is not reflected in the pictures.

1.         There are errors in the format of text, such as “withour” (line 267), mirna (line 357). Please carefully check the text format, grammar, etc. to improve language quality.

Reviewer 2 Report

This is a wonderful interesting article made at the highest research level. 1) 2.1. Sequencing quality analysis. For this section, I would recommend presenting the table in text rather than as a supplementary. 2) Figure 2, I would recommend splitting it into several pictures, which will increase the font and improve its perception 3) The primer sequences used in the study must be provided. In the form of supplementary. 4) Figure 3B - no scale bar. 5) It would be great if the authors could confirm the results of PCR analysis using Western blotting 6) The discussion is redundant, or needs to be structured.

Reviewer 3 Report

The authors have investigated the roles and mechanisms of T-2 toxin and selenium deficiency involved in Kaschin-Beck disease via high-throughput RNA sequencing technology using a rat model. This topic is interesting. The experiment was well-designed. The following revision can improve the quality of the paper.

1. Please briefly introduce the animal experimental design in the abstract, including the animal. sample size, selenium and T-2 toxins doses etc.

2. L37, please add the recently reference of the toxicity of T-2 toxins. Such as 1)MiR-214-3p may alleviate T-2 toxin-induced chondrocyte apoptosis and matrix degradation by regulating NF-κB signaling pathway in vitro, Toxicon, 2023; 2) T-2 toxin-induced intestinal damage with dysregulation of metabolism, redox homeostasis, inflammation, and apoptosis in chicks,Archives of Toxicology,2023ï¼›3) The response of glandular gastric transcriptome to T-2 toxin in chicks. Food and Chemical Toxicology. 2019.

3. L 48,selenium deficiency can induce multiple tissue damage and cause relative disese, pleasestrengthen the description of this point; and the recently reference such as: Selenium deficiency-induced multiple tissue damage with dysregulation of immune and redox homeostasis in broiler chicks under heat stress, Science China Life Sciences, 2023.

4. Please add the replicates n=? in the figure legends.

5. L131, please correct 'p-value' to 'p' and checking the simialr issues throughout the paper. 

6. Is there any histology and biochemistry data to indicate the damage of the articular cartilage, bone marrow and skeletal muscle etc? 

7. Please disscuse the findings with the simialr reshearch. such as: MiR-140 is involved in T-2 toxin-induced matrix degradation of articular cartilage, Toxicon, 2023.

8. L298-306, please add the rational for the doses chsoen dor SeMet and T-2 toxin. 

9. Please provided the concentration of Selenium and T-2 toxin in the basal diet.

Round 2

Reviewer 2 Report

The article can be accepted for publication in its current form

Reviewer 3 Report

The authors have revised the paper followed the comments. No further comments.